# Metabolomic Signatures in Doxorubicin-Induced Metabolites Characterization, Metabolic Inhibition, and Signaling Pathway Mechanisms in Colon Cancer HCT116 Cells

**DOI:** 10.3390/metabo12111047

**Published:** 2022-10-31

**Authors:** Raja Ganesan, Vasantha-Srinivasan Prabhakaran, Abilash Valsala Gopalakrishnan

**Affiliations:** 1Institute for Liver and Digestive Diseases, College of Medicine, Hallym University, Chuncheon 24253, Korea; 2Department of Biological Sciences, Pusan National University, Busan 46241, Korea; 3Department of Bioinformatics, Saveetha School of Engineering, Saveetha Institute of Medical and Technical Sciences (SIMATS), Chennai 602105, India; 4Department of Biomedical Sciences, School of Biosciences and Technology, Vellore Institute of Technology (VIT), Vellore 632014, India

**Keywords:** HCT116 cells, doxorubicin, colorectal cancer, NMR, metabolomics, metabolic profiling

## Abstract

Doxorubicin (DOX) is a chemotherapeutic agent is used for various cancer cells. To characterize the chemical structural components and metabolic inhibition, we applied a DOX to HCT116 colon cancer cells using an independent metabolites profiling approach. Chemical metabolomics has been involved in the new drug delivery systems. Metabolomics profiling of DOX-applied HCT116 colon cancer cellular metabolisms is rare. We used ^1^H nuclear magnetic resonance (NMR) spectroscopy in this study to clarify how DOX exposure affected HCT116 colon cancer cells. Metabolomics profiling in HCT116 cells detects 50 metabolites. Tracking metabolites can reveal pathway activities. HCT116 colon cancer cells were evenly treated with different concentrations of DOX for 24 h. The endogenous metabolites were identified by comparison with healthy cells. We found that acetate, glucose, glutamate, glutamine, sn-glycero-3-phosphocholine, valine, methionine, and isoleucine were increased. Metabolic expression of alanine, choline, fumarate, taurine, o-phosphocholine, inosine, lysine, and phenylalanine was decreased in HCT116 cancer cells. The metabolic phenotypic expression is markedly altered during a high dose of DOX. It is the first time that there is a metabolite pool and phenotypic expression in colon cancer cells. Targeting the DOX-metabolite axis may be a novel strategy for improving the curative effect of DOX-based therapy for colon cancer cells. These methods facilitate the routine metabolomic analysis of cancer cells.

## 1. Introduction

Colorectal carcinoma is the world’s third most commonly diagnosed cancer and the second most common cause of death from cancer [1]. Chemotherapeutic drugs of doxorubicin hydrochloride (DOX) have been utilized to eradicate leukemia cells due to their anticancer activity and inhibition of tumorigenesis by reducing cell division. However, DOX-based drug resistance has limited their utility [2,3]. The development of novel cancer therapeutics relies heavily on in vivo systems. At the same time, cancer cell lines provide a comprehensive dataset of the alterations in the metabolite levels in cancer cells and can produce significant information [4]. DOX is used as a chemotherapeutic drug for many cancers. Clinicians frequently need to reduce the dose, reducing efficacy [5,6].

Metabolomics studies have already shown alterations in colon cancer metabolites [7,8,9,10]. Metabolomics is typically performed using millions of cells, cultured cells, and tumor specimens [11,12]. Metabolomics has been a great study for comprehensive small-molecule analysis. The small molecules (e.g., sugars, amino acids, proteins, nucleic acids, and lipids) are essential for improving the overall quality of disease and are required for palliative care [13]. As a result, metabolites serve as the strongest predictors of cellular phenotypic expression and reveal substantial changes inside cancer cells [14,15]. Additionally, metabolites can interact with other biomolecules expressed in sick cells. Today, NMR spectroscopy’s metabolomic profiling in cancer cells is a rapidly advancing technology.

Herein, we describe an evaluating biomimetic pattern in colorectal carcinoma (HCT116 cancer cells) using metabolomics analysis that was proposed for investigating the DOX-applied metabolic pathways in cells. We also evaluate how the DOX treatment of HCT116 cancer cells affected their metabolic pathways. The experimental analyses in this work utilize cytotoxicity and metabolomics approaches. Herein, we demonstrate a practical approach to DOX-applied cellular viability, metabolic pathways, and metabolic discrimination in colon cancer cells. We provide strong evidence that cancer metabolic phenotypes reflect DOX-associated systemic metabolic changes. 

## 2. Materials and Methods

### 2.1. Chemicals and Reagents

Doxorubicin hydrochloride (DOX; Product Number: D1515; CAS Number: 25316-40-9; formula: C_27_H_29_NO_11_.HCL) was obtained from Sigma–Aldrich Chemicals (St. Louis, MO, USA). DOX has a molecular weight of 579.98 g/mol and is in the form of powder. Methanol (CH_3_OH) and chloroform (CHCl_3_) were purchased from Sigma–Aldrich. Dulbecco’s Modified Eagle’s medium (DMEM; Lot No: 09221712, GenDEPOT, St. Louis, MI, USA) and 10% (*v*/*v*) fetal bovine serum (FBS; Lot No: WB0009) were purchased from GE Healthcare Life Science (Queensland, Australia). HCT116 colon cancer cells were purchased from American Type Culture Collection (ATCC; Manassas, VA, USA). TSP-d_4_ is a 3-(trimethylsilyl) propionic acid that was purchased from Cambridge Isotope Laboratories in the United States.

### 2.2. DOX Solution Preparation

DOX was distributed in an aqueous media in accordance with earlier findings [16,17]. Briefly, 0.75, 1.5, and 7.5 µg/mL of DOX was added to 1% (*w*/*v*) bovine serum albumin (BSA) in 2 mL phosphate buffer solution (PBS). In order to improve the dispersibility of DOX, BSA was utilized. The obtained suspensions were then dispersed using a homogenizer with 200 W for 10–15 min under sterile conditions. After centrifugation at 2000× *g* for 15 min, insoluble DOX aggregates in the pellets were removed, and the supernatants were further homogenized by ultrasonic dispersive technology at 200 W for 15 min under sterile conditions. Cells were homogenized using a Precellys24 (PeqLab, Erlangen, Germany). The resulting DOX dispersions were kept at 4 °C for further research. Each solution was made with Milli-Q water with a resistivity (ultrapure water: 18.2 MΩ·cm).

### 2.3. Cell Culture and DOX Treatments

HCT116 colorectal cancer cells (HCT116 cells) (5 × 10^6^ cells per well) were cultured in DMEM supplemented with 10% (*v*/*v*) FBS and 1% (*v*/*v*) penicillin/streptomycin. The cells we employed in this work were passaged 2 to 5 times and kept in DMEM at 37 °C with 5% CO_2_ at subconfluence conditions.

Furthermore, HCT116 cells were incubated for at least 24 h with L-glutamine (1%, 2.0 mm), FBS (10%), penicillin (100 U/mL), streptomycin (100 µg/mL), and gentamycin (1 mL) at 5% CO_2_ incubator in 37 °C. Cells were stained with trypan blue dye (0.04%) once cell confluency was attained, and hemocytometer measurements were made. DOX was sonicated for 15 min before addition. HCT116 cells were evenly divided into four different concentrations (Control, 0; 0.75, 1.5, and 7.5 µg/mL DOX; *n* = 6) and incubated for 24 h at 37 °C.

### 2.4. WST-8 Cell Viability Assay

According to previous tests [18], HCT116 cells were seeded in 96-well plates at 5 × 10^3^ cells per well, leaving them to adhere for 24 h. Then culture medium was changed by DOX different concentrations for 24 h. After incubation, highly sensitive water-soluble tetrazolium salt (WST-8; 10 μL/mL) was added to each well, and cell cytotoxicity was measured using a Cellrix^®^ viability assay kit (Seoul, Korea). Cell viability was calculated by normalizing the absorbance at 450 nm. The non-DOX-treated cells were considered to be 100% viable.

### 2.5. Isolation of HCT116 Cells and Quenching Metabolome

Samples were prepared for metabolomics analysis as previously described [16,19,20]. First, 2 mL of medium for each sample was collected after 24 h, frozen in liquid nitrogen, and stored at −80 °C. All samples were allowed to thaw for 5 min. Next, 80 µL of chloroform was added to the samples at 4 °C and left to incubate for 30 min. Samples were then homogenized by vortexing and transferred to a 1.5 mL plastic tube. Next, 125 µL of deionized water and 125 µL of chloroform were added at 4 °C and vortexed again. For phase separation, samples were centrifuged at 13,000× *g* for 20 min at 4 °C. Subsequently, the two phases found in the aqueous phase (top: polar metabolites) and the organic phase (bottom: nonpolar metabolites) were separated into different tubes. The aqueous phase (polar metabolites) was lyophilized overnight to remove methanol and water.

A lot of research was conducted on the outcomes of cell quenching in liquid nitrogen. This typically serves as the initial step in various extraction techniques (immediately following the removal of the growing medium) right before cell washing to remove the medium residues. However, it was found in this study that washing the DOX-applied HCT116 cells after the quenching stage resulted in a considerable loss of the cell metabolites, most likely because the freezing step causes the cell wall to collapse, resulting in metabolite leakage. Quenching and washing were performed in reverse order to solve this issue.

Samples were allowed to thaw for 5 min. Then, 400 µL of all samples was diluted with 100 µL of buffer solution to a final concentration of 150 mM with phosphate buffer (pH: 7.2 in 100% D_2_O) containing 1 mM TSP-d_4_ as an internal standard. This solution was then added to the aqueous phase. After being vortexed and centrifuged at 12,000× *g* for 5 min, the supernatant from the samples was transferred in the amount of 550 µL to a 5 mm NMR tube.

### 2.6. Instrumentation of NMR and Data Files

NMR data were analyzed as previously described [16,21]. Briefly, 1D proton (^1^H)-NMR spectra were recorded at 26 °C using a Bruker Advance 600 MHz NMR spectrometer (Billerica, MA, USA). Spin-echo sequence (recycle delay-90°-(τ-180°-τ) n -acquisition) of t_2_- edited Carr–Purcell–Meiboom–Gill (CPMG) was used. Water-suppressed ^1^H-NMR spectra were acquired. Before measurements, for equilibrium maintenance, samples were kept for 5 min inside the NMR probe at 298 K. A total of 128 scans were requested. The width was found at 9615.4 Hz. ^1^H-spectral resonance width contained 15 to −1 ppm, and each sample took 8–10 min to be measured. The resonance of the water domain (δ 4.48–4.68) was removed. The phosphate buffer containing TSP-d4 was used as a reference at 0 ppm. The data settings of ^1^H (e.g., baseline, filling, and TSP adjustment at 0 ppm) were handled by using the Chenomx NMR Suite software (Chenomx Inc., Edmonton, AB, Canada).

### 2.7. Computational Tools and Statistical Analysis

Use of the in-house MetaboAnalyst 5.0 software, score plot analysis such as principal component analysis (PCA), and orthogonal projections to latent structures discriminant analysis (OPLS-DA) were applied to the NMR data. The goodness of fit (R^2^), the goodness of prediction (Q^2^) of the score plot, metabolite sets enrichment analysis (MSEA), and VIP scores (cutoff, >1) were assessed. The difference between R^2^ and Q^2^ decreases with the score plots’ robustness.

The logarithmic transformation of the acquired concentrations was used to reduce the impact of both noise and the high variance of the variables. *Pareto scaling* was applied. It can divide each variable by the square root of its standard deviation. The normalized metabolite concentrations (mM) are presented as the mean ± standard deviations (SD). *p*-values < 0.05 were considered to be statistically significant.

## 3. Results

### 3.1. Optimization of the Quenching and Extraction Procedures for DOX-Treated Cells

Figure 1a shows the chemical structure of DOX [22]. As per Martineau et al. [23], ^1^H-NMR-based metabolomics analysis to recover the cell metabolites was achieved by combining different steps. Identifying several crucial portions that needed a significant amount of optimization was possible by examining and assessing the various extraction techniques. Figure 1b shows the flowchart design and the experimental pipeline for the metabolomics analysis. All data were autoscaled before analysis. The spectral features were extracted and used for pathway analysis in extracted metabolites.

### 3.2. Cell Viability-Treatment with DOX in HCT116 Cells

The cell viability of HCT116 cells cultured in the *hypoxic* environment decreased by 81 percent from 6 to 72 h. The effect of treatments with DOX and cellular viability are summarized in Figure 2.

DOX-applied HCT116 cells develop into a subconfluent monolayer, and it was discovered that it was particularly challenging to remove the cell growth media during the washing process entirely. Because of its residual signals that were dispersed throughout the central area of the NMR spectrum, glucose, one of the most prevalent components of the medium, presented special difficulties for the spectral interpretation of the extracted metabolome. There are now four washing steps instead of three to prevent this. After cleaning, the cells could be mechanically scraped off the plates, and the metabolic activity of the cells could then be quickly put to rest by liquid nitrogen.

### 3.3. Integration of Targeted Metabolomics Datasets

This study determined and targeted 50 metabolites with DOX concentrations using ^1^H-edited spectroscopic peaks. Most 1H signals fall in the 0.5–5.5 ppm region, so we focused the analysis on it. Phosphate buffer was employed as a reference. Per *Pascal’s triangle*, the spectral data multiplicity of DOX was calculated and calibrated. The potential metabolites were profiled and quantified using their ^1^H nuclei (e.g., CH, CH_2_, and CH_3_) and proton multiplicity, as shown in Table 1.

This technique is utilized to estimate probability density functions, which may result in practical applications of probability distributions in DOX-applied HCT116 cells. The metabolites and their normalized concentrations (mM) were evaluated based on the solvent. OPLS-DA score plot-based DOX in HCT116 cellular metabolic discrimination are as follows: 0.75 µg/mL, 58.9%; 1.5 µg/mL, 75.8%; and 7.5 µg/mL, 39.5%. These data could disperse the fundamental level of metabolic variations (Figure 3a–c). The unsupervised technique of PCA score plot was used to visualize basic metabolic clarity in DOX-treated HCT116 cell data (data not shown). At 1.5 µg/mL of DOX, the highest percentage of metabolic classification was found compared to others. The OPLS-DA score plot indicated that all samples were within the 95% confidence interval (Hotelling T2 ellipse). OPLS-DA analysis was used to classify and filter the metabolites.

The cumulative R^2^ and Q^2^ values were greater than 0.6, indicating that the OPLS-DA score plot was suitable for determining the difference between control and HCT116 cells (Figure 3d). The metabolic deference was screened from low DOX to high DOX concentrations. This metabolic difference of DOX treatment was further confirmed with a 100-premutation test, which could verify its stability and reliability. Here, the OPLS-DA score plot did not overfit.

### 3.4. Volcano Plots and Heatmap Analysis of HCT116 Cells Metabolites

Figure 4a–c show volcano plots of differentially expressed metabolites in HCT116 cancer cells induced by DOX. False discovery rates (FDRs) were plotted against DOX-induced log_2_ fold changes in HCT116 cancer cells. The 1.5 µg/mL of DOX-applied up- and downregulated metabolites were found to be more than 0.75 µg/mL of DOX. Interestingly, there are no upregulated metabolites in 7.5 µg/mL of DOX-treated in HCT116 cells (Figure 4c). The DOX-applied volcano plot has been studied in all drug-applied groups. A volcano plot of HCT116 cellular metabolites in positive and negative modes was summarized.

The DOX-treated top 25 metabolites’ phenotypic expression is shown as a heatmap in Figure 4d. The metabolites from heatmap analysis of HCT116 cells were identified as candidate biomarkers. The metabolites were filtered and collated with a VIP score > 1 (data not shown). The top 25 metabolites were identified as candidate biometers. Expression levels were assessed and compared to determine the metabolites in 0.75 µg/mL, 1.5 µg/mL, and 7.5 µg/mL of DOX in HCT116 cells.

### 3.5. Tailored Metabolic Adaptations to DOX and Primary Metabolites with Pathways

Among the amino acid metabolites, glutamate, glutamine, sn-glycero-3-PCho, valine, methionine, and isoleucine were increased in HCT116 cells by DOX treatment. Acetate and glucose were increased in cancer cells (Figure 5a). Acetate and glucose are the by-products of energy metabolism. DOX concurrently increased many significant metabolites in HCT116 cells. Several amino acids were significantly decreased by DOX, when amino acids may be involved in autophagy and mitophagy [24]. DOX treatment also induced metabolic downregulation in specific amino acids such as alanine, choline, taurine, o-phosphocholine, inosine, lysine, and phenylalanine (Figure 5b). These observations suggested a decrease in ATP production through glycolysis and oxidative phosphorylation. This could be related to the inhibition of metabolic synthesis in the cells. 

The metabolites were quantified as before by targeted metabolomics [25]. The increase in hepatic glucose content suggests an enhancement of glycogenesis. Taurine and glutathione (GSH) were most closely associated with liver injury after DOX exposure. GSH, which is biosynthesized from cysteine, glycine, and glutamate, is the most abundant cellular redox molecule and plays a significant role in antioxidant functions [15,19].

The DOX effect appears specific for neurological metabolites, glutamate, and glutamine passed to induce neurogenesis behaviors. The neurological effects of glutamate and glutamine were observed in adult mice C57BL/6J mice treated with indole and DOX [26]. Several molecules are also associated with neurological dysfunction. The relative concentration of eight upregulated and eight downregulated metabolites from the ^1^H-NMR analysis were analyzed (Figure 5). These metabolites showed valid metabolites for OPLS-DA modal discrimination. There are many significant metabolites among DOX-treated groups.

Phosphocholine is a metabolite that partially comes from processing dietary carnitine into trimethylamine and choline. The evidence demonstrated that valine was associated with brain injury, which our study further confirmed by increased valine in DOC treatment. L-valine deficiency is noticeable in neurological defects in the brain [27].

The top metabolites identified by this analysis highlighted changes in selected metabolic pathways in HCT116 cancer cells. Pathways enrichment analysis was employed using the total quantified metabolites (Figure 6a–c). The pathways are represented as circles. We confirmed that metabolites taken from cancer cells undergoing transformation demonstrated significant changes in metabolites and metabolic pathways. Taurine and purine metabolism were highly altered in cancer cells, which could be used at therapeutic intervals. This greater capacity is further confirmed by HCT116 cells having a higher mitochondrial volume than control or DOX cells.

The metabolites regulation successfully distinguished and identified the cancer metabolites from the control metabolites. The gut microbe ecosystem modulates and shapes many metabolic, immunological, structural, and neurological functions [28]. Most appropriate metabolisms were marked.

Figure 6 shows that 0.75 µg/mL of DOX significantly regulated the citrate cycle (TCA cycle); taurine and hypotaurine metabolism; valine, leucine, and isoleucine degradation; and alanine, aspartate, and glutamate metabolism. At 1.5 µg/mL of DOX in colon cancer cells, the alanine, aspartate, and glutamate metabolism; D-glutamine, and D-glutamate metabolisms; citrate cycle (TCA cycle); and taurine and hypotaurine metabolism have been altered. Similarly, arginine and proline metabolism; alanine, aspartate, glutamate metabolism; D-glutamine and D-glutamate metabolism; and glycolysis/gluconeogenesis most robustly contributed to 7.5 µg/mL of DOX treatment.

Study findings support that changes in colon cancer metabolites could contribute to physiological responses to gut microbiota. Human prostate cancer metabolomics via high-resolution NMR spectroscopy has analyzed the metabolites in prostate tissue, seminal fluid, serum, and urine [29,30]. Figure 7 represents the HCT116 cancer cells after DOX treatment. The schematic diagram of glycolysis and TCA cycle imbalance by DOX in colon cancer cells has been summarized.

This study has some limitations. These findings show that DOX resistance exhibits separate basic metabolic vulnerabilities despite shared chemotherapeutic adaptations. Metabolic therapies can address these vulnerabilities to inhibit drug-resistant tumor growth.

## 4. Discussion

The study revealed the metabolite regulation of HCT116 cells with DOX treatment. We reduced the noise ratio in all spectrums. We increased the spectral signal resolution by employing NMR spectroscopy. We reduced contamination by eliminating sample drying and increased NMR performance by extracting metabolites with 80% acetonitrile [31,32]. This technique can be used to evaluate any cellular metabolites, but in practice, it is most helpful when cell counts are modest and enzymatic dissociation is unnecessary. The R^2^Y and Q^2^Y values in OPLS-DA score plots were observed in the distinct metabolic clusters of DOX-treated HCT116 cells [33,34]. 

Our observation of increased glucose levels by DOX treatment might suggest the inhibition of glycolytic processes and could be linked with the inhibition of cell proliferation. The intervention in glutamate metabolism by glutaminase inhibitors was shown to impact glutathione levels, which could reduce the antioxidative tumor capacity. Due to the high incidence of DOX-mediated anticancer activity, research usually positively affects cellar proliferation. DOX-responsive metabolic profiling detected early toxicity metabolite signature [35]. Evidence suggests that DOX causes these current harmful consequences by producing free radicals, lipid peroxidation, and oxidative damage to tissues and organs [36,37].

An increased amount of acetate and glucose in cancer cells was observed after repeated exposure to DOX. It was concluded that the DOX might remarkably block some signaling pathways. The alteration in glycolysis, TCA cycle, nucleotide synthesis, and choline metabolisms could be associated with growth suppression, which we found in DOX-treated HCT116 cells. These findings underscore the significance of colon cancer metabolites that regulate metabolic reactions that disturb energy, amino acids during metabolic transcriptional changes in mitochondria, and adult neurotransmitter impairment. Herein, we inferred the mechanism of metabolites inhibition enhancing the efficacy of chemotherapeutics could be clarified through metabolomics analysis. Notably, the changes in metabolites indicated that DOX was likely a source of therapeutic drugs for colon cancer cells. The DOX also contains other bioactive therapeutic drug roles.

Breast cancer cells resistant to both medications demonstrated well-known anthracyclines resistance mechanisms in a cancer cell line, such as enhanced drug efflux, lysosomal activity, and oxidative stress response [38,39]. Our in vitro studies display direct, metabolite-specific disorders, demonstrating phenotypic expression [40,41]. 

Our data gave evidence that DOX inhibits the TCA cycle-medicated mitochondrial damage. However, the reduced fumarate production by DOX may inhibit the ATP molecule production from TCA cycle transporters. Our study shows TCA cycle arrest, metabolite depletion, and mitochondrial dysfunction induced by DOX. In this context, impaired metabolites in mitochondria have been linked to anticancer activity [40,42,43]. 

We observed a metabolic difference in the abundance of metabolites in HCT116 cells. Functional studies will be required to assess the biological significance of the difference. Cancer cells are undergone metabolic changes to survive oxidative stress during metastasis [44,45]. A better understanding of the metabolic changes could reveal new therapeutic vulnerabilities to colon cancer progression.

## 5. Conclusions

We have implemented a fully automated study in HCT116 cancer cells with optimized peak detection, alignment, and an annotation task using ^1^H-NMR data. The raw spectra processing produces a peak intensity table and peak annotation table. We found that the metabolites of glutamate, glutamine, sn-glycero-3-PCho, valine, methionine, and isoleucine were increased. The specific amino acids such as alanine, choline, taurine, o-phosphocholine, inosine, lysine, and phenylalanine were downregulated in colon cancer HCT116 cells. In the future, we aim to screen more data in HCT116 cells by DOX treatment to support cancer metabolic regulation.

## Figures and Tables

**Figure 1 metabolites-12-01047-f001:**
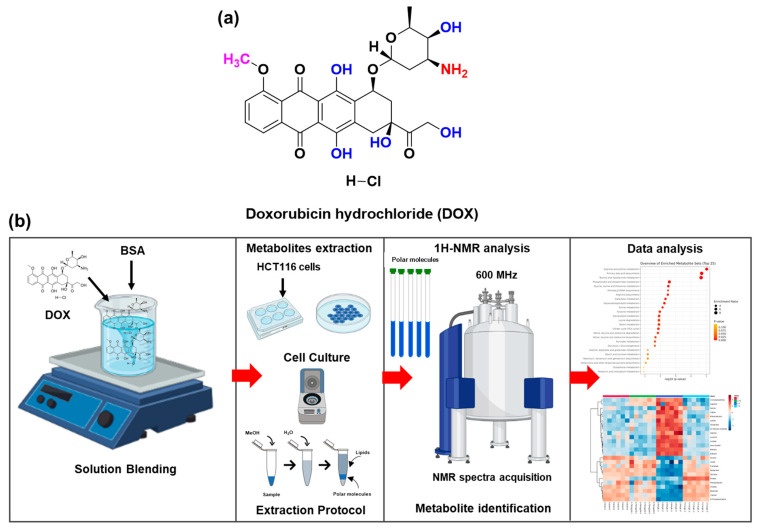
The schematic depicts the current HCT116 cells utilized in metabolomics and its future metabolic therapeutic applications. (**a**) Chemical structure of doxorubicin hydrochloride. (**b**) The experimental flow of DOX-applied HCT 116 cells and ^1^H-NMR-based metabolomics analysis.

**Figure 2 metabolites-12-01047-f002:**
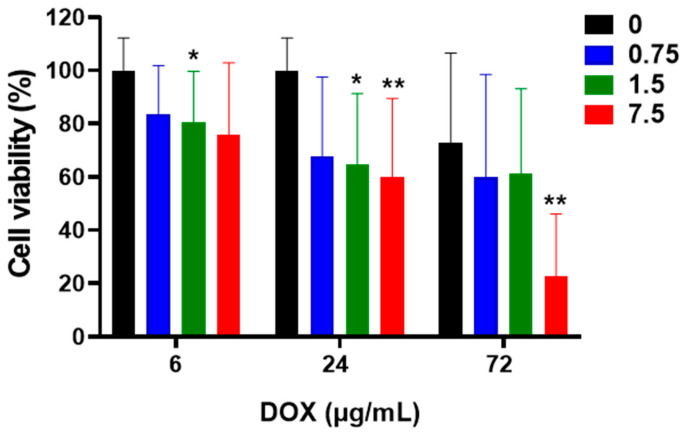
DOX and colon cancer cell viability affect different times (6, 24, and 72 h). Three DOX concentration (0, control; 0.75 µg/mL; 1.5 µg/mL; and 7.5 µg/mL) at three time points were recorded. At least 6 replicates were used for each tested concentration. The bar graphs show 0, black; 0.75 µg/mL, blue; 1.5 µg/mL, green; and 7.5 µg/mL, red. Tested with 2-way ANOVA with Tukey’s multiple comparison test, significant adjusted *p* < 0.05. * and ** represent *p* < 0.05 and *p* < 0.001, respectively.

**Figure 3 metabolites-12-01047-f003:**
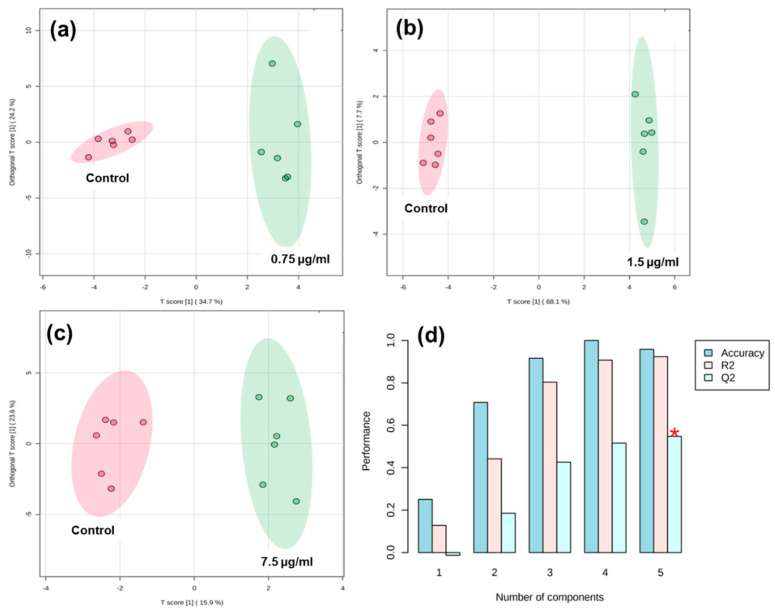
Outline of HCT116 cancer metabolic profile based on OPLS-DA scores scatter plot visualization. Results from the raw spectra processing. (**a**) 0.75 µg/mL of DOX; (**b**) 1.5 µg/mL of DOX; (**c**) 7.5 µg/mL of DOX-treated HCT116 cells. The pink and green circles indicate control and DOX-treated cancer cells, respectively. Two-Dimensional OPLS-DA score plot in HCT116 cancer metabolic profile visualization. (**d**) R^2^ and Q^2^ performance analysis. A red asterisk indicates the highest expression of Q^2^ values of components.

**Figure 4 metabolites-12-01047-f004:**
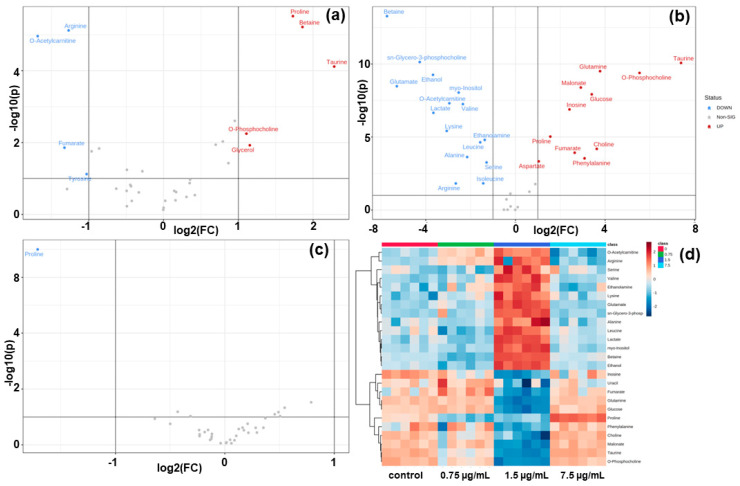
Functional volcano plot and heatmap analysis of quantified metabolome patterns: (**a**) 0.75 µg/mL DOX; (**b**) 1.5 µg/mL DOX; (**c**) 7.5 µg/mL DOX treated in HCT116 cells. This is a screenshot of a heatmap based functional analysis in HCT116 cells. (**d**) The correlation matrix of the top 25 metabolites significantly altered. The correlation type (positive or negative) and strength (color intensity) are coded brown and blue, respectively, normalized between −3 and +3 according to the bar on the right.

**Figure 5 metabolites-12-01047-f005:**
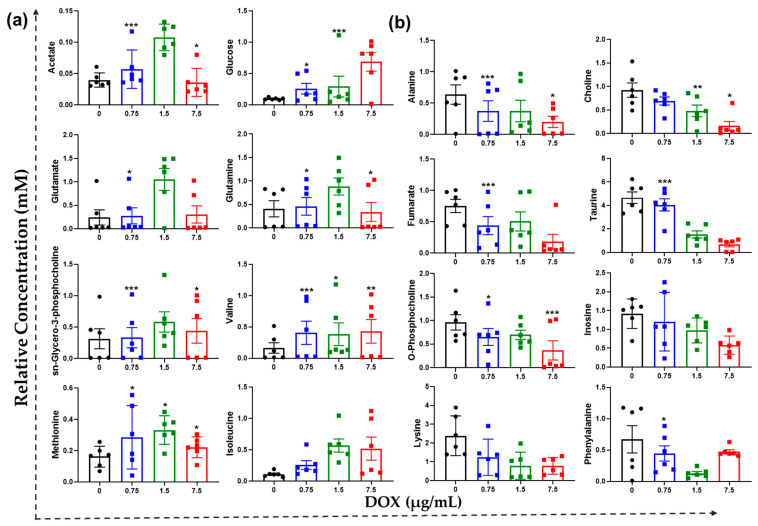
Essential metabolites are assigned higher priority scores using metabolomics profiling with DOX treatment. (**a**) Expression of acetate, glucose, glutamate, glutamine, sn-glycero-3-PCho, valine, methionine, and isoleucine were increased metabolites in HCT116 cancer cells. (**b**) Expression of alanine, choline, fumarate, taurine, o-phosphocholine, inosine, lysine, and phenylalanine were decreased their regulation in HCT116 cancer cells. The black dot represents the control; the blue dot represents 0.75 µg/mL of DOX; the green dot represents 1.5 µg/mL of DOX; and the red dot represents 7.5 µg/mL of DOX. *, ** and *** represent *p* < 0.05, *p* < 0.01, and *p* < 0.001, respectively.

**Figure 6 metabolites-12-01047-f006:**
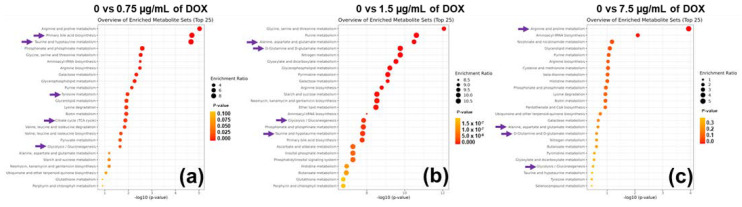
(**a**) 0.75 µg/mL of DOX; (**b**) 1.5 µg/mL of DOX; (**c**) 7.5 µg/mL of DOX. The metabolite set enrichment analysis of our metabolites with DOX treatment in HCT116 cells. Pathway analysis is displayed from HCT116 cells datasets using the Holm-Bonferroni method and False Discovery Rate. Each row represents a pathway, and each column represents a -log10 (*p*-value). The topmost pathway shows the results from the top down (according to the enrichment ratio). A purple arrow indicates significant metabolic pathways.

**Figure 7 metabolites-12-01047-f007:**
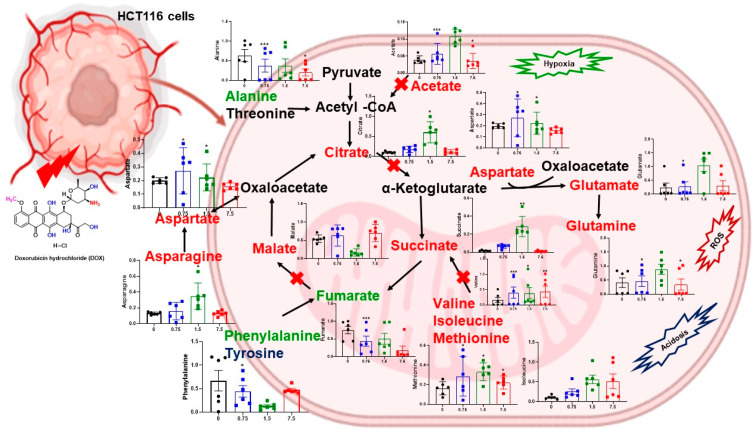
The schematic regulation of TCA cycle-associated activity by DOX treatment in HCT116 cancer cells. Red- and green-colored metabolites were represented as increased and decreased in HCT116 cells, respectively. The red sign shows a possible metabolic chemical reaction inhibition in cancer cells. *, ** and *** represent *p* < 0.05, *p* < 0.01, and *p* < 0.001, respectively.

**Table 1 metabolites-12-01047-t001:** The characterization of metabolome by ^1^H-NMR spectroscopy. ^1^H-NMR-based metabolomic quantification and targeted metabolites from cancer cells.

δ ^1^H (ppm) and Multiplicity ^a^	Targeted Metabolites	Chemical Formula	MW (Da)
1.90 (s)	Acetate	C_2_H_4_O_2_	60.05
3.89 (dd); 2.80 (dd); 2.66 (dd)	Aspartate	C_4_H_7_NO_4_	133.10
3.78 (q); 1.47 (d)	Alanine	C_3_H_7_NO_2_	89.09
3.76 (t); 1.90 (m); 1.68 (m)	Asparagine	C_6_H_14_N_4_O_2_	132.12
8.52 (s); 8.12 (d); 4.50 (m); 4.21 (m)	ATP	C_10_H_16_N_5_O_13_P_3_	507.18
8.54 (s); 5.94 (m); 4.11 (m); 4.00 (m)	ADP	C_10_H_15_N_5_O_10_P_2_	427.201
8.22 (s); 6.16 (s); 4.53 (dd); 4.34 (d)	AMP	C_10_H_14_N_5_O_7_P	347.221
3.89 (s); 3.25 (s)	Betaine	C_24_H_26_N_2_O_13_	550.45
2.65 (d); 2.53 (d)	Citrate	C_6_H_8_O_7_	192.12
4.05 (dd); 3.50 (dd); 3.18 (s)	Choline	C_5_H_14_NO	104.17
6.51 (s)	Fumarate	C_4_H_4_O_4_	116.072
5.22 (d); 4.64 (d); 3.88 (dd); 3.72 (m); 3.40 (m)	Glucose	C_6_H_12_O_6_	180.16
4.20 (q); 3.78 (m); 2.97 (dd); 2.15 (m)	Glutathione	C_10_H_17_N_3_O_6_S	307.32
3.76 (t); 2.44 (m); 2.12 (m)	Glutamate	C_5_H_9_NO_4_	147.129
5.75 (m); 7.80 (m); 6.16 (t)	Glutamine	C_5_H_10_N_2_O_3_	146.144
4.10 (q); 1.32 (d)	Lactate	C_3_H_6_O_3_	342.3
3.66 (d); 1.96 (m); 0.99 (d); 0.92 (t)	Isoleucine	C_6_H_13_NO_2_	131.17
3.60 (d); 2.261 (m); 0.97 (d)	Valine	C_5_H_11_NO_2_	117.146
3.72 (m); 1.70 (m); 0.94 (t)	Leucine	C_6_H_13_NO_2_	131.17
8.30 (s); 6.05 (d); 4.42 (dd); 3.82 (dd)	Inosine	C_10_H_12_N_4_O_5_	268.23
2.37 (s)	Oxalacetate	C4H4O5	132.071
2.61 (m); 2.51 (dd); 2.11 (dd); 1.07 (d)	Methylsuccinate	C_5_H_8_O_4_	132.11
4.05 (t); 3.61 (t); 3.52 (dd); 3.26 (t)	Myo-Inositol	C_6_H_12_O_6_	180.16
7.33 (d); 7.37 (m); 7.43 (m)	Phenylalanine	C_9_H_11_NO_2_	165.19
2.46 (s)	Pyruvate	C_3_H_4_O_3_	88.0621
2.39 (s)	Succinate	C_4_H_6_O_4_	118.088
3.43 (t); 3.42 (t); 3.25 (t)	Taurine	C_2_H_7_NO_3_S	125.15
4.24 (m); 3.57 (d); 1.31 (d)	Threonine	C_4_H_9_NO_3_	119.12
7.72 (d); 7.31 (s); 4.04 (dd); 3.29 (dd)	Tryptophan	C_11_H_12_N_2_O_2_	204.26
2.89 (s)	Trimethylamine	C_3_H_9_N	59.1103
7.24 (d); 6.94 (m); 3.34 (dd); 3.30 (dd)	Tyrosine	C_9_H_11_NO_3_	181.188
3.25 (s)	TMAO	C_3_H_9_NO	75.11
7.892 (s)	Xanthine	C_5_H_4_N_4_O_2_	152.110

Note and Abbreviations; ^a^—Singlet (s); doublet (d); triplet (t); doublet of doublets (dd); quartets (q); multiplets (m); molecular weight (MW); parts-per-million (ppm); adenosine monophosphate (AMP); adenosine diphosphate (ADP); adenosine triphosphate (ATP); Trimethylamine N-oxide (TMAO).

## Data Availability

The data presented in this study are available in the article.

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
