# Peer review of "Metabolomic Signatures in Doxorubicin-Induced Metabolites Characterization, Metabolic Inhibition, and Signaling Pathway Mechanisms in Colon Cancer HCT116 Cells"

_metabolites, 2022, doi:10.3390/metabo12111047_

Round 1

Reviewer 1 Report

In the article “Metabolomic Signatures in Doxorubicin-Induced Colon Cancer HCT116 Cells and Metabolites as Novel Anti-Cancer Drugs”, the authors aimed to investigate the metabolic profile of colon cancer cells treated with the chemotherapeutic Doxorubicin (DOX). The authors explored NMR for this analysis and evaluated the metabolome of cells exposed to increasing concentrations of DOX. According to the authors, the level of a series of metabolites were altered by DOX treatments.

Despite the effort made by the authors, this reviewer believes that this manuscript requires a complete revision and restructuring to warrant publication in MDPI Metabolites. This recommendation is based on the following:

1) The manuscript is confusing and does not communicate a clear message. For example, in the second paragraph of introduction, the authors say that “metabolomics-based colon cancer studies are reported”. In the following paragraph, however, they state “using metabolomics analysis by DOX therapy, there are no established methodologies”. Why this drug treatment invalidates the existing method? Does it mean every new treatment will require a different protocol for metabolites quantification? This is unrealistic.

2) How this claimed new method compares to classic LC-MS analysis? What are its advantages? And disadvantages?

3) A lot of details regarding the methodology are spread in the results section. The authors should revise all this information and make sure they are confined into the appropriate section.

4) The results section has several disconnected sentences, which does not contribute to the overall message. For example, the study targets colon cancer but states that “The liver stores lipids before releasing them into the circulation [25]. Therefore, changes in HCT116 metabolites indicate hepatic dysfunction.” What the authors are trying to say? Please, revise.

5) In the same section is stated that “Similar neurogenetic effects were also observed in adult mice C57BL/6J mice treated with indole”. What is the relevance of this to the manuscript?

6) How the data obtained compares to public dataset of colon cancer’s metabolome? 

7) The whole manuscript must be revised to correct grammar and spelling mistakes. 

Overall, this reviewer recommends the full revision of the manuscript by the authors in order to increase its quality and improve its message, prior to resubmitting to this or any other journal.

Author Response

# Reviewer 1:

Comments and Suggestions for Authors

In the article "Metabolomic Signatures in Doxorubicin-Induced Colon Cancer HCT116 Cells and Metabolites as Novel Anti-Cancer Drugs", the authors aimed to investigate the metabolic profile of colon cancer cells treated with the chemotherapeutic Doxorubicin (DOX). The authors explored NMR for this analysis and evaluated the metabolome of cells exposed to increasing concentrations of DOX. According to the authors, the level of a series of metabolites was altered by DOX treatments.

Response: We are grateful for the reviewer's valuable comments. Many thanks for your scientific comments and appreciation of our research work. The goal of this research article is to understand only the changes in metabolomics profile following DOX exposure. In response to your concerns, we used DOX at various concentrations and the metabolite regulation found in colon cancer cells.

Despite the effort made by the authors, this reviewer believes that this manuscript requires a complete revision and restructuring to warrant publication in MDPI Metabolites. This recommendation is based on the following:

  • The manuscript is confusing and does not communicate a clear message. For example, in the second paragraph of the introduction, the authors say that "metabolomics-based colon cancer studies are reported ."In the following paragraph, however, they state, "using metabolomics analysis by DOX therapy, there are no established methodologies ." Why this drug treatment invalidates the existing method? Does it mean every new treatment requires a different metabolite quantification protocol? This is unrealistic.

Response: We apologize for the unclear sentences. It is now corrected in the revised manuscript. Different cancer studies are reported based on metabolomics. We worked with 600 MHz NMR spectra of DOX-applied cancer cells and a pipeline of methodologies to drive metabolite changes.

  • How did this claim new method compares to classic LC-MS analysis? What are its advantages? And disadvantages?

Response: High-throughput analytical methods, including LC/GC-MS and NMR applications, have been frequently applied to investigate and quantify the metabolites in given samples. Since NMR is found to be more accurate in metabolomics studies and our laboratory possesses adequate expertise in studying the alterations in metabolites using NMR, we have opted for NMR over LC/MS analysis.

  • A lot of details regarding the methodology are spread in the results section. The authors should revise all this information and make sure they are confined to the appropriate section.

Response: As per the reviewer's points, we have modified it according to your instructions. The basic points of this study are discussed. We feel this information could be important to understand the complete metabolomic works.

  • The results section has several disconnected sentences, which do not contribute to the overall message. For example, the study targets colon cancer but states, "The liver stores lipids before releasing them into circulation [25]. Therefore, changes in HCT116 metabolites indicate hepatic dysfunction." What are the authors trying to say? Please, revise.

Response: It is now corrected in the revised manuscript. We are sorry for the unclear sentence.

  • The same section states that "Similar neurogenetic effects were also observed in adult mice C57BL/6J mice treated with indole". What is the relevance of this to the manuscript?

Response: The sentence was corrected. Neurological metabolites were studied with DOX exposure. 

  • How was the data obtained compared to a public dataset of colon cancer's metabolome?

Response: The raw 1H-NMR data were obtained for each assayed platform of 1H-NMR chemical shifts. Database metabolites were initially annotated using a public dataset (The Human Metabolome Database (HMDB: https://hmdb.ca/). Blank matric samples were used as control cells. Colon cancer HCT116 cells were compared with cellular control metabolites.

  • The whole manuscript must be revised to correct grammar and spelling mistakes. Overall, this reviewer recommends the full revision of the manuscript by the authors to increase its quality and improve its message before resubmitting it to this or any other journal.

Response: We have done an official language edition before submitting it to the journal. Thanks for your comments.

Reviewer 2 Report

The manuscript by Ganesan et al reports metabolomic signatures of doxorubincin (DOX)-treated colon cancer cells. The authors applied 1H-NMR spectroscopy to determine changes in metabolome of HCT116 cells treated with different concentrations of DOX and analyzed the results with various statistical methods. The authors determined metabolites increased and decreased upon DOX treatments and identified pathways associated with the metabolites. The authors also claimed that they applied new protocol to improve specificity of metabolome analysis. The manuscript provides novel information on metabolome of DOX-treated colon cancer cells, which merits publication. However, there are a few points that are not described clearly and that conclusions are deduced without proper experimental supports.

1.      ‘~ Metabolites as Novel Anti-Cancer Drugs’ in Title is not supported by the results.  

2.      Confusing sentences in ‘Abstract’

-          In lines 15~16, ‘Colon cancer cells that have been exposed to DOX are becoming more and more understood to contain therapeutic compounds that control phenotypic expression.’

-          In Lines 25~26, ‘While improving the DOX concentration, the metabolic phenotypic expression in colon cancer steroids fluctuated.’

3.      In Section 2.2, information on the homogenizer is missing. What is the final concentration of DOX after removal of DOX aggregate?

4.      Lines 100 and 103: Is ‘BOX’ DOX?

5.      ‘sample was collected from each concentration around 2 mL after 24 h’ in line 106: What does this mean?

6.      Line 169, ‘the hypoxic environment’. Is this correct?

7.      Line 175, ‘Turkey’s multiple’. Turkey or Tukey?

8.      Lines 200, the mentioned data is advised to be included as a supplemental data.

9.      Line 247, ‘glutamate, glutamine, glutamate, glutamine, sn-‘

10.  Line 255~256, ‘5b)\, ‘Using existing hypotheses, we further extrapolated from these data the total rates of ATP production through glycolysis and oxidative phosphorylation might be altered.’ Please specify the alteration.

11.  Lines 260 and 272, Is ‘DOC’ DOX?

12.  Lines 289-290, ‘This methodology proved successful in distinguishing and identifying the exogenous metabolites from the endogenous metabolites.’ How exactly is the exogenous metabolites distinguished from the endogenous ones?

13.  Figure 4 and 6, the axis labels are illegible. Bigger and clear labels are required.

14.  Lines 331~336. How is this paragraph related with current study?

15.  Lines 337~339: Please specify the changes related with ‘~ improvements in biomarkers of cancer cellular function.’ Meanwhile, the first sentence of this paragraph seems awkward.

16.  Lines 340~343, ‘behavior that disturbance’ and ‘adult neurogenesis impairment’ Need to expand discussion for each category in more detail.

17.  Lines 343~345: Need to provide grounds for this notion and to explain how the metabolites could be likely source of therapeutic drugs in more detail.

18.  Lines 348~349: What is the ground for this claim?

19.  Lines 350~353: How is this paragraph related with current study?

20.  Line 361, ‘This this study~’

21.  Possible cause of concentration-dependent difference in metabolite profile mentioned in lines 226~230 is required to be discussed.

22.  What are the benefits of new protocol employed in this study?

Author Response

# Reviewer 2:

Comments and Suggestions for Authors

The manuscript by Ganesan et al. reports metabolomic signatures of doxorubicin (DOX)-treated colon cancer cells. The authors applied 1H-NMR spectroscopy to determine changes in the metabolome of HCT116 cells treated with different concentrations of DOX and analyzed the results with various statistical methods. The author's determined metabolites increased and decreased upon DOX treatments and identified pathways associated with the metabolites. The authors also claimed that they applied a new protocol to improve the specificity of metabolome analysis. The manuscript provides novel information on the metabolome of DOX-treated colon cancer cells, which merits publication. However, a few points are not described clearly, and the conclusion is deduced without proper experimental support.

Response: Many thanks for your scientific comments and appreciation of our research works. As per your scientific comments, we have revised the manuscript prepared without damaging the structure, shape, and quality of science in this manuscript.  

'~ Metabolites as Novel Anti-Cancer Drugs' in the Title is not supported by the results. 

Response: we revised the title (new title: Metabolomic Signatures in Doxorubicin-Induced Metabolites Characterization, Metabolic Inhibition, and Signaling Pathway Mechanisms in Colon Cancer HCT116 Cells)

  1. Confusing sentences in 'Abstract.'

-          In line 15~16, 'Colon cancer cells that have been exposed to DOX are becoming more and more understood to contain therapeutic compounds that control phenotypic expression.'

-          In Lines 25~26, ‘While improving the DOX concentration, the metabolic phenotypic expression in colon cancer steroids fluctuated.’

Response: We apologize for the unclear sentence. It is now corrected in the revised manuscript.

  1. In Section 2.2, information on the homogenizer is missing. What is the final concentration of DOX after the removal of DOX aggregate?

Response: Cells were homogenized using a Precellys24 (PeqLab, Erlangen, Germany). Metabolites were expressed in mM (relative concentration). The 0.75, 1.5, and 7.5 µg/ml DOX were used for HCT116 cells.

  1. Lines 100 and 103: Is ‘BOX’ DOX?

Response: It is now corrected in the revised manuscript. We are sorry for the typo error.

  1. ‘sample was collected from each concentration around 2 mL after 24 h’ in line 106: What does this mean?

Response: We edited the sentence and explained it in a transparent way.

  1. Line 169, ‘the hypoxic environment .'Is this correct?

Response: Yes. We corrected in manuscript. 

  1. Line 175, ‘Turkey’s multiple’. Turkey or Tukey?

Response: Tukey is correct. We revised the manuscript. Thank you.

  1. In line 200, the mentioned data is advised to be included as supplemental data.

Response: We corrected in manuscript. 

  1. Line 247, ‘glutamate, glutamine, glutamate, glutamine,sn-‘

Response: We corrected the metabolite repetition. Sorry for the typo error.

  1. Line 255~256, '5b)\, 'Using existing hypotheses, we further extrapolated from these data the total rates of ATP production through glycolysis, and oxidative phosphorylation might be altered.' Please specify the alteration.

Response: We have completely revised the sentence in the manuscript. These observations suggested a decrease in ATP production through glycolysis and oxidative phosphorylation. This could be related to the inhibition of metabolic synthesis in the cells.

  1. Lines 260 and 272, Is ‘DOC’ DOX?

Response:  We corrected in manuscript. We are sorry for the unclear word.

  1. Lines 289-290, 'This methodology successfully distinguished and identified the exogenous metabolites from the endogenous metabolites.' How exactly are the exogenous metabolites distinguished from the endogenous ones?

Response:  We have completely revised the sentence in the manuscript.

  1. In Figures 4 and 6, the axis labels are illegible. More prominent and clear labels are required.

Response: We improved the axis labels as much as possible. These figures are computational (metal analyst data).

  1. Lines 331~336. How is this paragraph related to the current study?

Response: We have completely revised the sentence in the manuscript.

  1. Lines 337~339: Please specify the changes related to '~ improvements in biomarkers of cancer cellular function.' Meanwhile, the first sentence of this paragraph seems awkward.

Response: We removed the sentences.

  1. Lines 340~343, 'behavior that disturbance' and 'adult neurogenesis impairment,' Need to expand the discussion for each category in more detail.

Response: We have completely revised the sentence in the manuscript.

  1. Lines 343~345: We need to provide grounds for this notion and explain how the metabolites could likely be a source of therapeutic drugs in more detail.

Response: We corrected in manuscript.

  1. Lines 348~349: What is the ground for this claim?

Response: We corrected in manuscript.

  1. Lines 350~353: How is this paragraph related to the current study?

Response: We corrected the paragraph in the manuscript.

  1. Line 361, ‘This thisstudy~’

Response: We modified the manuscript.

  1. The possible cause of the concentration-dependent difference in the metabolite profile mentioned in lines 226~230 must be discussed.

Response: Yes. The metabolite profile in each concentration-dependent difference is important. 

  1. What are the benefits of the new protocol employed in this study?

Response: The benefits of this study are quantified and targeted the metabolites regulation in DOX-applied HCT116 cancer cells. This could be a fundamental analysis for therapeutic biomarker invention.

Round 2

Reviewer 1 Report

Following the corrections made by the authors, this reviewer suggests the acceptance of the manuscript.